# An Evaluation Approach of Community Emergency Management Ability Based on Cone-ANP

**DOI:** 10.3390/ijerph20032351

**Published:** 2023-01-28

**Authors:** Kai Wang, Yuanyuan Feng, Jun Deng, Chang Su, Quanfang Li

**Affiliations:** 1College of Safety Science and Engineering, Xi’an University of Science and Technology, Xi’an 710054, China; 2Xi’an Key Laboratory of Urban Public Safety and Fire Rescue, Xi’an 710054, China

**Keywords:** community emergency management, index system, Cone-ANP, weight, membership degree

## Abstract

In order to improve the emergency management ability of urban communities in response to emergencies and reduce the losses caused by emergencies, based on the method of Cone-Analytic Network Process (Cone-ANP), a whole-process community emergency management ability evaluation method was proposed. Using around 25 evaluation indexes from six dimensions, namely infrastructure resilience, community organization resilience, risk management, emergency material support, emergency force building, and emergency literacy, this method established the dominant relationship of each index by the analysis of the cone network structure. It determined the community safety culture construction as the cone-top element, and obtained the limit weight vector of all the evaluation indexes by expert evaluation. The membership degree of each index was calculated to quantify the evaluation results of community emergency management ability. The results could provide a guidance and reference basis for community emergency management.

## 1. Introduction

The concept of “community” was coined by Ferdinand Tonnies to describe “close-knit, helpful, and humane social groups of homogeneous populations of shared values [1,2].” Community is a community formed by interrelated people in a certain field and its activity area. It is the most basic place to maintain the lives of residents and the first place where unexpected accidents occur. Community emergency management is an important part of maintaining community safety. Community emergency management mainly refers to the ways and means to deal with any emergency the first time, so that residents can avoid or reduce losses [3,4].

With the continuous development of cities, community emergency management capacity has become an important indicator to measure the management capacity of each city and social civilization. In recent years, China’s emergency management has made historic achievements and undergone historic changes, and the Chinese characteristic emergency management system of “all kinds of disasters, big emergencies” has basically taken shape [5]. However, due to the short time and weak foundation of China’s emergency management system reform, the current effect is not significant, especially in the community emergency management; there are still a series of problems, such as imperfect community working mechanisms, insufficient arrangement of community emergency management personnel, insufficient objectives and accurate evaluation of community emergency management, and inadequate methods for assessing community emergency response capacity. Therefore, it is of great practical significance to explore a set of effective and scientific evaluation methods for responding to public health emergencies to improve community emergency capacity building.

Scholars’ research on emergency management mainly focuses on CAR evaluation system [6,7]; through the emergency management theory, it further develops comprehensive emergency management, and improves and expands the evaluation index system obtained. Among them, Mann believes that the US government lacks response measures in the face of biochemical attacks and should further strengthen the emergency management of biochemical attacks to obtain relevant measures [8,9]; Adini put forward the pyramid theory on emergency response capability. He believed that the government’s plans and policies, infrastructure and facilities, personnel’s knowledge reserves and capabilities, and emergency training exercises can form a relatively complete emergency response capability [10,11]; Daniel et al. established an emergency management indicator system based on the government’s four aspects of emergency prevention, emergency preparedness, emergency response, and emergency recovery [12,13,14,15].

In addition, at present, scholars mainly use AHP, fuzzy evaluation, network analysis, particle swarm optimization, and the artificial neural network, etc [16,17,18,19,20], when establishing the emergency management index system. The evaluation effect is good; however, there are still some defects, such as the uncertainty in the evaluation not being fully considered, and the construction of the emergency response capacity evaluation system being unreasonable. The analysis hierarchy method is a commonly used analysis method; in order to overcome that AHP cannot reflect the interrelationship between elements in complex systems, Saaty [21,22] proposed ANP on the basis of AHP, further considering the dependence and feedback relationship between element sets. However, according to this method, Wu, Yu, and Tzemg [23,24] believe that ANP has theoretical defects in analyzing the dependencies within the element set, and it is difficult for decision makers to judge the dependence and feedback relationship between element sets. Yu and Tzemg also use fuzzy cognitive graph techniques to indirectly describe the relationship between dependence and feedback [25]; however, they do not substantially address this shortcoming. In addition, Saaty [21] only stops at the concept of ANP, and does not deeply describe or describe the system structure characteristics of evaluation and decision-making problems. From previous studies on the indicator system, it is found that the internal elements of the indicator system are complex and numerous, and their mutual dominance relationship is complex, which will inevitably lead to multiple cone-top elements and special cone-bottom elements of an „only entering but not leaving” type; there are some limitations in the research [26,27,28].

In view of the theoretical shortcomings of the above methods and the need to solve complex evaluation and decision-measurement problems, this paper adopts a new analytical structure different from ANP [29], namely the method of Cone-Analytic Network Process (Cone-ANP); further, solving the problems of uncertainty in the evaluation that is not fully considered, and the unreasonable construction of the emergency response capacity evaluation system. In this paper, the emergency management index system is studied by using the cone network analysis method, paying more attention to distinguishing the dominance relationship between each element, clearly describing it, better distinguishing the cone-top element and the cone-bottom element, analyzing the system structure of the community emergency management index system based on the cone network analysis method; and finding the weight of each index, ranking according to the weight, proposing countermeasures to reduce the casualties and property losses caused by emergencies.

In recent years, Cone-ANP has been mainly used for pipeline risk assessment, and systematically constructs a pipeline integrity risk assessment system, including design, construction [30], operation, and decommissioning stages from the perspective of the whole life cycle of oil and gas pipelines. The evaluation and study of the emergency capacity of power transmission and transformation projects improve the emergency response capacity level of power transmission and transformation projects, and reduce the losses caused by accidents. In this paper, Cone-ANP is used for the evaluation of community emergency management ability.

## 2. Construction of Evaluation Index System for Community Emergency Management

After reviewing a large amount of literature, starting from the whole process of community emergencies for the four stages of emergency prevention, emergency preparedness, emergency response and emergency recovery [31,32], six dimensions (secondary indicators) of infrastructure, community organization, risk and hidden danger management, emergency material support, emergency force construction, and emergency literacy were preliminarily screened. By visiting and investigating community managers, community residents, merchants, and other personnel, combined with the actual situation of the community and the external environmental factors (community service capacity, community infrastructure construction, community safety culture construction, community management ability, etc.) and their own emergency response capabilities (education, safety awareness, psychological conditions, etc.) were analyzed. Experts in related fields and scholars with high experience were invited to discuss according to the “National Comprehensive Disaster Reduction Model Community Standards” document issued by the National Disaster Reduction Commission [33,34], and 25 three-level indicators were obtained, named e1~e25 [35], as shown in Figure 1. For this indicator system, the analysis of Cone-ANP can fully reflect the structural relationship of each element within the system, and it can better understand which links are weak in the community and need to be further improved.

## 3. Cone-ANP Structure of Emergency Assessment Indicators

### 3.1. Divide the Taper Element Set

According to the dominant relationship among the 25 cone elements, they are divided into cone-top elements and cone-bottom elements. The elements at the top of the cone can dominate the elements at the bottom of the cone and are not dominated by any elements at the bottom of the cone, while the elements at the bottom of the cone can dominate each other. The structural characteristics of the cone-top element and the cone-bottom element in the figure are: the cone bottom element can point to other cone-bottom elements, but not to the cone-top element; that is, the arrows between the cone-bottom elements can be bidirectional; the cone-top element can only point to the cone-bottom element and its arrows are unidirectional.

Among them, the cone-base elements can be divided into two types: one is the “only in but not out” type, which is expressed as all rows are 0 and not all columns are 0 in the matrix, indicating that the element can only be dominated by other elements in the element set and not other elements, also known as “acceptability” cone-base elements; the other type is the “both in and out” type, which is represented in the matrix as the row and column not being zero, indicating that this element can be dominated by other cone-base and cone-top elements in the element set, and can also dominate other cone-base elements, which is called the “transitional” cone-base element. The cone-top elements are represented in the matrix as “all columns are 0, and not all rows are 0”. They are “only in but not out” elements, also known as “originating” cone-top elements. In this paper, the community emergency evaluation index is divided into a cone element set $C=\{c_{i}|i=1,2,3,\cdots ,25\}$.

### 3.2. Dominant Relationship among Indicator Elements of Community Emergency Assessment

Invite experts to judge the dominant relationship between the community emergency evaluation indicator elements in Figure 2, and further construct the relationship matrix E. In the obtained relational matrix E, 1 indicates that ei has a dominant relationship with ej, and 0 indicates that ei does not have a dominant relationship with ej. In the relational matrix E, if the 25th column is 0 and the row is not 0, it means that e25 is a cone-top element and is not dominated by other elements, but can dominate other elements. The other 24 elements have one row and one column; that is, the “both in and out” type cone-bottom element; the community safety culture construction is the cone-top element; and the project earthquake resistance and disaster prevention capacity, emergency shelter, and other 24 elements except the community safety culture construction are all cone-bottom elements.

### 3.3. Community Emergency Management Index Cone-ANP Structure

According to the judgment results of the cone-top and cone-bottom elements of the community emergency management indicator system, combined with the mutual dominance relationship between the various emergency management indicators in matrix E, the analysis framework of the community emergency management indicator cone network is established, as shown in Figure 3.

## 4. Weight Calculation of Community Emergency Management Indicators

On the basis of the community emergency management index Cone-ANP, calculate the weight value of each index according to the following steps:

Step 1: Experts were invited to judge the relative importance of the cone-top and cone-bottom elements in Figure 3 according to the set of community emergency management indicator cone elements, and the judgment matrix was constructed using the 1–9 scale method in ANP. Then, the ANP method is used to calculate the relative weight of each cone-bottom element in the cone element set C relative to the cone-top element e25:$${(\delta_{1},\delta_{2},\delta_{3},\cdots \delta_{24})}^{Τ}=(0.01232,0.0659,0.0644,0,0,0.0457,0.0889,0.08585,0,0,0.1014,0,\;0.056,0.056,0.0538,0.0501,0.0413,0.0288,0.0288,0,0.0267,0.0267,0.0808,0.013{9)}^{Τ}.$$

Step 2: According to the relationship between each cone-base element and the cone-top element in matrix E, several experts were invited to judge the internal dominance relationship of all the cone-base elements and calculate the relative weight according to the pairwise judgment method of ANP relative weight, expressed as $\partial \left(e_{m*h_{m}*},e_{mh_{m}}\right)$, where $e_{mh_{m}}$ represents different elements, $e_{m*h_{m}*}$ represents the pairwise comparison preference weight assigned to different elements; among $m,m^{*}=1,h,h^{*}=1,2,3,\cdots 24,$ it is required that $h_{m}$ and $h_{m}*$ are not equal. For one element, use the AHP method to compare the importance of all the other cone-base elements in pairs; and then, obtain the relative weight $\partial \left(e_{m*h_{m}*},e_{mh_{m}}\right)$, if two elements do not dominate each other, $\partial \left(e_{m*h_{m}*},e_{mh_{m}}\right)=0$. According to the weight distribution principle $\sum_{m^{*}=1}\sum_{h_{m}*\neq h_{m}}\partial \left(e_{m*h_{m}*},e_{mh_{m}}\right)=1$, according to this, matrix B can be constructed, as shown in Figure 4:

Step 3: In this Cone-ANP structure, when t = 0, 1, 2, the weight of all the elements in the bottom of element set C is represented by ω, respectively, expressed as ${(\omega_{1}^{\left(t\right)},\omega_{2}^{\left(t\right)},\cdots \omega {}_{24}^{\left(t\right)})}^{Τ}$; according to the basic principle of the network analysis method, we can obtain:(1)$$\omega_{25}^{\left(t\right)}=\omega_{1}^{\left(t\right)}+\omega_{2}^{\left(t\right)}+\omega_{3}^{\left(t\right)}+\cdots \omega_{24}^{\left(t\right)}$$
(2)$$\omega_{h_{m}}^{\left(t\right)}=\delta_{h_{m}}\omega_{25}^{\left(t\right)},h_{m}=1,2,3,\cdots 24$$
(3)$$\omega_{h_{m}}^{\left(t\right)}=\delta_{h_{m}}\omega_{1}^{\left(t\right)}+\delta_{h_{m}}\omega_{2}^{\left(t\right)}+\cdots \delta_{h_{m}}\omega_{24}^{\left(t\right)},h\cdots =1,2,3\cdots 24$$
where $\omega_{i}^{\left(t\right)}$ is the weight of all the elements at the bottom of the base-cone in element set C, and $\delta_{h_{m}}$ is the composite weight.

Re order $W={(\omega_{1}^{\left(t\right)},\omega_{2}^{\left(t\right)},\cdots \omega {}_{24}^{\left(t\right)})}^{Τ}$. The sum of the weights of $W^{\left(0\right)}$ is 1, and all the weights are positive:(4)$$W^{\left(t-1\right)}=DW^{\left(t-1\right)}$$
(5)$$D={\left[\begin{array}{cccc} \delta_{1} & \delta_{1} & \cdots & \delta_{1}\\ \delta_{2} & \delta_{2} & \cdots & \delta_{2}\\ \vdots & \vdots & & \vdots \\ \delta_{24} & \delta_{24} & \cdots & \delta_{24} \end{array}\right]}_{24\times 24}$$
where $\delta_{i}$ is the relative weight.

According to Formula (4), when each cone-base element is known, at time t-1, discuss the influence of the internal and external dominance relationship of the remaining cone-base elements on each cone-base element in this Cone-ANP, and draw the weight at time t:(6)$$\omega_{h_{m}}^{\left(t\right)}=\sum_{1}^{24}\omega_{k}^{\left(t-1\right)}\alpha (e_{h_{m}},e_{k}),\;h_{m}=1,2,3,\cdots ,24,\;t=1,2$$

Therefore, we can get:(7)$$W^{\left(t\right)}=B\cdot W^{\left(t-1\right)},t=1,2\cdots$$

According to Formula (4), we can get:(8)$$W^{\left(t\right)}=B\cdot DW^{\left(t-1\right)}$$

To sum up, the product of matrix B and D is a random matrix, and according to the weight principle of the AHP method, the ownership weights are all positive numbers and the sum is 1. Then, let B•D=Q, we can obtain:(9)$$W^{\left(t\right)}=QW^{\left(t-1\right)}=Q^{2}W^{\left(t-2\right)}=Q^{3}W^{\left(t-3\right)}=\cdots Q^{t}W^{\left(0\right)}$$

According to Formula (9), if $Q^{\left(+\propto \right)}$ exists, then $W^{\left(+\propto \right)}$ exists, and must converge to $Q^{\left(+\propto \right)}$, not related to $W^{\left(0\right)}$; on the contrary, if $Q^{\left(+\propto \right)}$ does not exist, $W^{\left(+\propto \right)}$ must oscillate and converge. Then, the limit sorting weight vector of each cone-base element can be calculated as:(10)$$W^{\left(+\propto \right)}=Q^{\left(+\propto \right)}$$

According to the above analysis, use Matlab software to calculate the limit sorting weight vector of the cone-bottom elements and then, combine the Formulas (1) and (2) to obtain the weight of the cone-top elements; in addition, finally obtain the weight vector of each index of this Cone-ANP:$$W={\left(\omega_{1},\omega_{2},\omega_{3},\cdots ,\omega_{25}\right)}^{Τ}=(0.0175,0.0175,0.0175,0.0175,0.025,0.025,0.035,0.005,0.01,0.005,\;0.005,0.02,0.02,0.03,0.075,0.015,0.04,0.02,0.015,0.03,0.015,0.03,0.01,0,0.5{)}^{Τ}.$$

## 5. Analysis on Evaluation Results of Emergency Indicators

### 5.1. Quantitative Evaluation Results


(1)Determination of the evaluation grade


In the process of community emergency management, according to foreign emergency response capacity assessment standards, quantitative rules for evaluation shall be established, which are mainly divided into four grades: excellent, good, general, and poor. A certain scoring range shall be set, as shown in Figure 5.


(2)Quantitative evaluation and scoring


Experts are invited to score according to the quantitative scoring rules. The scoring system is a 10-point system. The average value of each expert’s scoring on each indicator is shown in Figure 6.

According to the scoring results, experts scored the lowest on emergency input, indicating that the community did not invest much in emergency response. Relevant leading departments and communities should increase emergency input to ensure basic emergency equipment and facilities.


(3)Analysis and evaluation results


After knowing the scoring of each indicator by each expert, quantify the evaluation results. The membership vector of each indicator is:$$\begin{array}{ll} \left(\omega_{1},\omega_{2},\omega_{3},\cdots ,\omega_{25}\right)= & (0.0175,0.0175,0.0175,0.0175,0.025,0.025,0.035,0.005,0.01,0.005,0.005,\\ & 0.02,0.02,0.03,0.075,0.015,0.04,0.02,0.015,0.03,0.015,0.03,0.01,0,0.5), \end{array}$$
expert evaluation quantitative score vector:$$\begin{array}{l} Y=(7.7,7.3,7.7,7,7.5,8,6.3,7.5,7,7,7.2,7.2,7,7.3,7.5,6.7,7.5,7,7.5,7.7,7.8,6.7,6.8,7.3,7.3).\\ P=WY^{Τ}=8.385 \end{array}$$

P is the total score of the indicator system, which also means that the community’s evaluation value of emergency response capacity is 8.385, belonging to the excellent level.

### 5.2. Analysis and Discussion

In this paper, the priority weight of community emergency management indicators is: safety culture construction, fire-fighting materials, the social reserve mechanism, emergency input, emergency materials reserve point, volunteer team, gap group, life passage, grid management, accident hidden danger management, disaster hidden danger management, social forces, engineering earthquake resistance and disaster prevention capacity, emergency shelter, community medical aid station, family reserve of community fire stations, disaster informants, emergency teams of enterprises and institutions, emergency response time, emergency plans and drills, organizational capacity, risk monitoring and early warning, urban lifeline risk prevention and control, and emergency science publicity and education activities. Then, the fuzzy comprehensive evaluation system is used in the evaluation of community emergency management ability, and the emergency response ability is finally excellent.

In community emergency management, no matter which link is wrong, any link error will lead to accidents and disaster [36]. From the evaluation of the community emergency management indicator system, it can be concluded that the construction of safety culture is the most important indicator, and the degree of membership of emergency science popularization and education activities is the smallest. However, it does not mean that the indicator of emergency science popularization and education activities is not important [37,38,39]. On the contrary, in the daily life of the community, science popularization and education activities are the most indispensable. Only through publicity activities can we make it clearer to everyone how to respond to emergencies. We will not be at a loss, and we can also arouse the idea of being emergency volunteers. Compared with the construction of safety culture, this indicator is more important in any field. Only the construction of community safety culture can make residents understand the construction of community safety, clarify their responsibilities, and know this well. This belongs to the infrastructure of a community [17].

Therefore, in community emergency management, we pay attention to weak links, such as emergency science popularization and education activities, risk monitoring and early warning, and urban lifeline risk prevention and control construction; further strengthening community emergency management and investment, and increasing emergency response time, so that we can respond calmly in the face of crisis. However, we cannot ignore the relatively good links we have established at present. We must take every link in the emergency management process seriously. Both managers and residents should have an emergency in mind; and have skills to implement in the face of emergencies, calmly respond, and reduce personal casualties and property losses caused by emergencies.

On the basis of the previous research, it is found that Cone-ANP has two significant advantages: (1) it no longer requires decision makers to subjectively judge and weigh the element set, and overcomes the arbitrary empowerment problem caused by the ANP’s block weighting method for element sets. The Cone-ANP in this paper no longer requires the subjective judgment of decision makers, significantly improves the scientific rationality of the evaluation conclusions, and provides new theoretical and technical support for community emergency management capabilities. (2) In terms of structure, the use of the pointed cone network analysis structure can not only fundamentally overcome the comparative logic of ANP in the construction of an element set self-dependency matrix, in the form of “comparing A and B relative to A”, but also reflect the difference in structural characteristics of element origin and transition that ANP does not reveal. Compared with ANP, Cone-ANP significantly improves the scientific nature of the evaluation conclusions. However, in addition, the disadvantage of Cone-ANP is that it is not possible to assess the emergency response capacity of the community as a whole, and new evaluation methods need to be constantly researched.

## 6. Conclusions


(1)In this paper, Cone-ANP is introduced into the evaluation of the community emergency management index system to avoid the randomness of subjective judgment. According to the index system proposed by experts, an element set is divided to determine the cone-top element and cone-bottom element in the element set. The cone-bottom element also includes receptivity and transition. The mutual dominance relationship between the elements is further determined. The general structure of the cone network analysis method and the calculation method of its weight are fully used to obtain the limit weight vector, and finally the membership degree of each index is calculated.(2)Based on Cone-ANP, community emergency management should strengthen emergency input, strengthen emergency science publicity and education activities, improve the emergency level of community residents, further increase risk monitoring and early warning, urban lifeline risk prevention and control, and improve community emergency management capacity.(3)Through the evaluation of community emergency management ability, it can provide an important reference value for the emergency management ability of community emergencies; and provide a basis for the emergency management work of urban management departments, further improve the level of community emergency management, and reduce the casualties and property losses caused by emergencies. It should be noted that the evaluation process of community emergency management capacity is complex and dynamic, and a single method cannot comprehensively assess the overall capacity of the community.(4)Community emergency management is a systematic and comprehensive work. Any link will lead to accidents. So, for the discovery of relatively weak links, managers must be timely according to the evaluation results, find out the shortcomings in emergency management work, and focus on strengthening.


## Figures and Tables

**Figure 1 ijerph-20-02351-f001:**
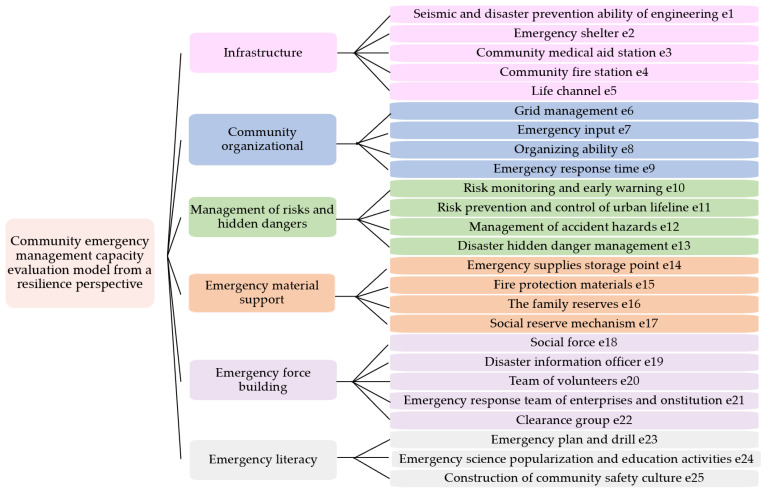
Community emergency management index system.

**Figure 2 ijerph-20-02351-f002:**
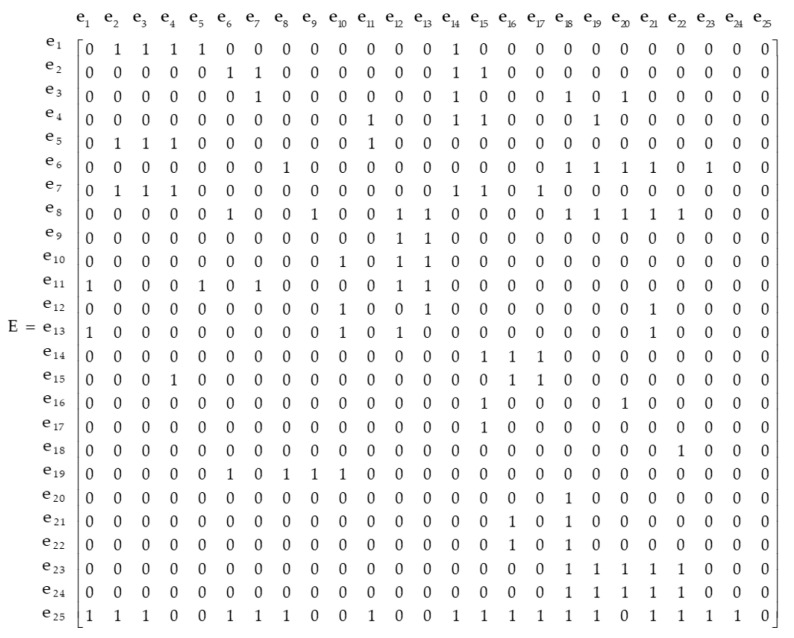
Dominant relationship matrix between indicator elements of community emergency assessment.

**Figure 3 ijerph-20-02351-f003:**
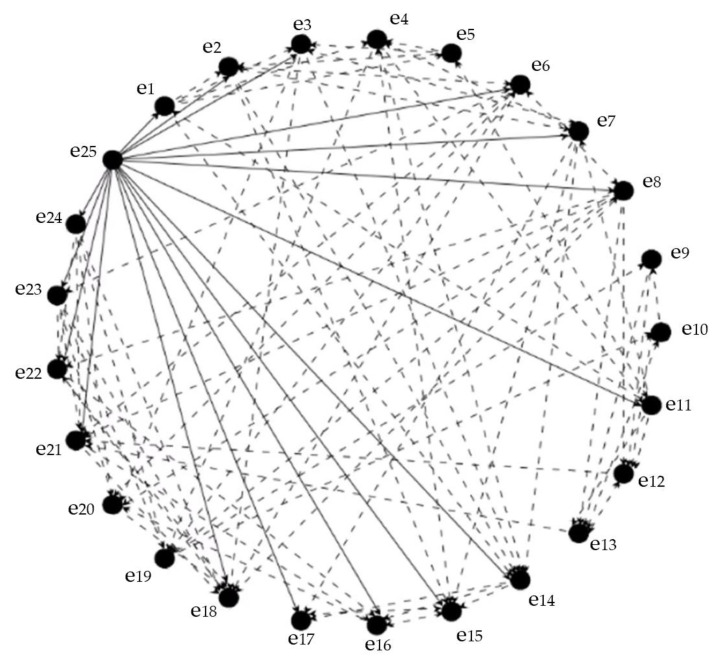
Community emergency management index pyramid network analysis structure chart.

**Figure 4 ijerph-20-02351-f004:**
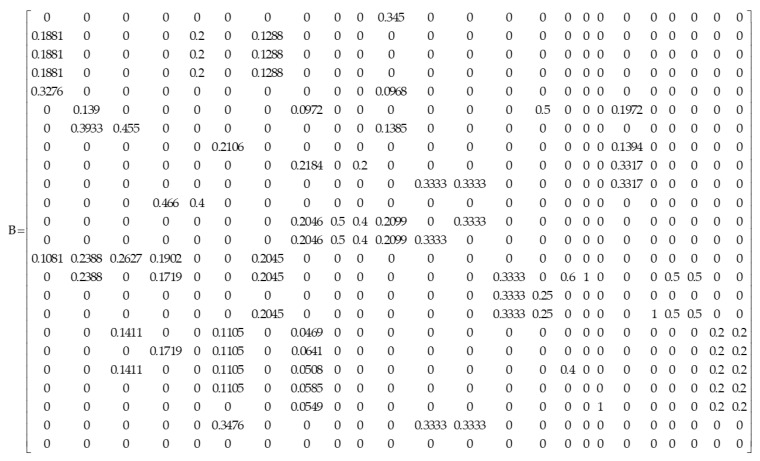
Weight relationship matrix between indicator elements of community emergency assessment.

**Figure 5 ijerph-20-02351-f005:**
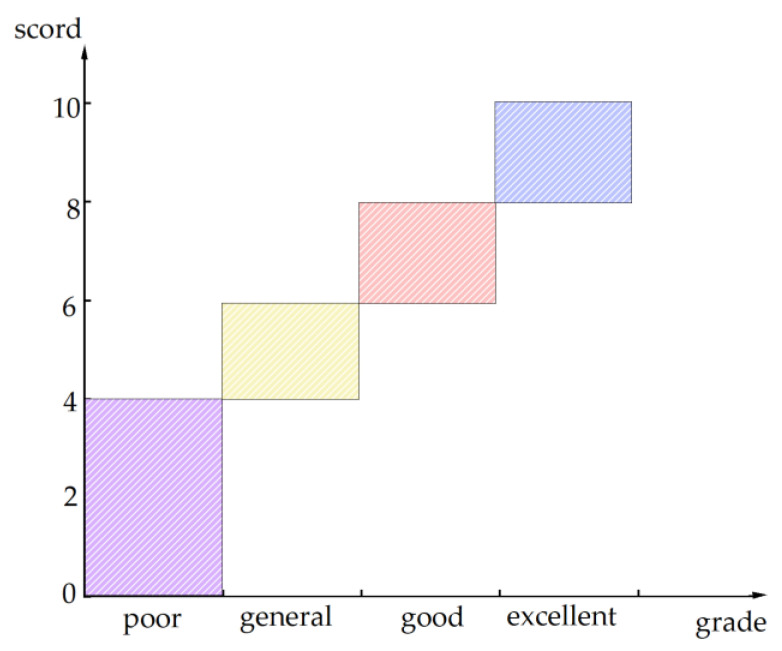
Evaluation quantification chart.

**Figure 6 ijerph-20-02351-f006:**
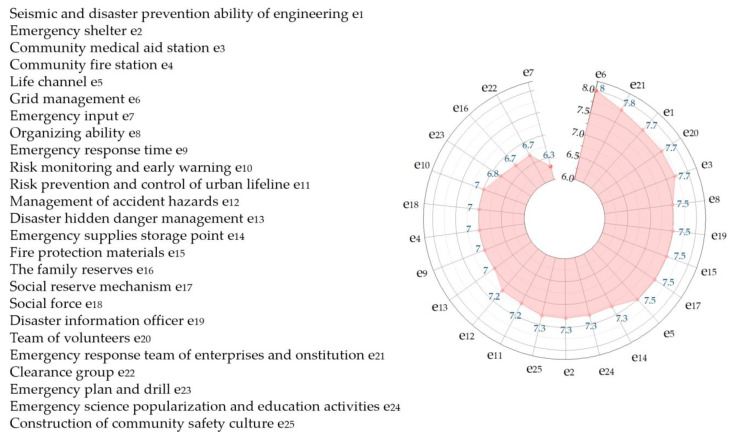
Specific scoring results of Level III indicators.

## Data Availability

The data that support the findings of this study are available from the corresponding author upon reasonable request.

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
