# Peer review of "An Evaluation Approach of Community Emergency Management Ability Based on Cone-ANP"

_ijerph, 2023, doi:10.3390/ijerph20032351_

Round 1

Reviewer 1 Report

Based on Cone-Analytic network process (CONE-ANP), this paper proposes a method for evaluating the capacity of community emergency management in the whole process. The leading relationship between the indicators is established by analyzing the cone network structure. Although the paper has done some work, there are several problems. I have the following criticisms.

1.  The Introduction and Literature Review are insufficient to support the novelty. The introduction lacks the discussion of the importance and necessity of this topic. It is necessary to give more detailed explanations on what this research adds and summarizing the main findings clearly.

2.  The second section lacks a relevant explanation for selecting the indicator system, so the reasons for choosing each indicator should be added to increase its persuasiveness.

3.  What is the innovation of this decision model? Is there any unexpected result in the research?

4.  The authors should try to present some punchline insight in conclusions and managerial suggestions

Reviewer 2 Report

This interesting paper aims to integrate ANP with cone network analysis. However, there are some critical issues that must be improved before considering for publication.

(1) Why did this study integrate cone with ANP? It is still unclear about this development. Can we apply only ANP or other adapted ANP approaches presented by other past works?

(2) The contribution of work mainly comes from the proposed method. Therefore, the discussion section should focus more on the implementation of the method. Trying to compare with the existing body of knowledge (similarity, and dissimilarity to other related works; advantages, and disadvantages of proposed method).

(3) Limitations and practical implication should be provided.

Round 2

Reviewer 1 Report

Overall, I am relatively satisfied with your modification

Reviewer 2 Report

All my comments have been properly addressed. The paper can be accepted for publication.